# Association of Increased Vascular Stiffness with Cardiovascular Death and Heart Failure Episodes Following Intervention on Symptomatic Degenerative Aortic Stenosis

**DOI:** 10.3390/jcm11082078

**Published:** 2022-04-07

**Authors:** Jakub Baran, Anna Kablak-Ziembicka, Pawel Kleczynski, Ottavio Alfieri, Łukasz Niewiara, Rafał Badacz, Piotr Pieniazek, Jacek Legutko, Krzysztof Zmudka, Tadeusz Przewlocki, Jakub Podolec

**Affiliations:** 1Department of Interventional Cardiology, Institute of Cardiology, Jagiellonian University Medical College, Prądnicka 80 Str., 31-202 Krakow, Poland; jakub_baran@yahoo.pl (J.B.); kablakziembicka@op.pl (A.K.-Z.); kleczu@interia.pl (P.K.); rbadacz@gmail.com (R.B.); jacek.legutko@uj.edu.pl (J.L.); zmudka@icloud.com (K.Z.); 2The John Paul II Hospital, Prądnicka 80 Str., 31-202 Krakow, Poland; lniewiara@gmail.com (Ł.N.); kardio@kki.krakow.pl (P.P.); tadeuszprzewlocki@op.pl (T.P.); 3Noninvasive Cardiovascular Laboratory, The John Paul II Hospital, Prądnicka 80 Str., 31-202 Krakow, Poland; 4San Raffaele University Hospital and Alfieri Heart Foundation, 20132 Milan, Italy; alfieri.ottavio@hsr.it; 5Department of Emergency Medicine, Faculty of Health Sciences, Jagiellonian University Medical College, 31-126 Krakow, Poland; 6Department of Cardiac and Vascular Diseases, Institute of Cardiology, Jagiellonian University Medical College, the John Paul II Hospital, Prądnicka 80 Str., 31-202 Krakow, Poland

**Keywords:** vascular stiffness, cardiovascular death, degenerative aortic stenosis, heat failure episodes, pulsatile index, resistive index, aortic valve replacement, transcatheter aortic valve implantation

## Abstract

Background. The resistive (RI) and pulsatile (PI) indices are markers of vascular stiffness (VS) which are associated with outcomes in patients with cardiovascular disease. We aimed to assess whether VS might predict incidence of cardiovascular death (CVD) and heart failure (HF) episodes following intervention on degenerative aortic valve stenosis (DAS). Methods. The distribution of increased VS (RI ≥ 0.7 and PI ≥ 1.3) from supra-aortic arteries was assessed in patients with symptomatic DAS who underwent aortic valve replacement (AVR, n = 127) or transcatheter aortic valve implantation (TAVI, n = 119). During a 3-year follow-up period (FU), incidences of composite endpoint (CVD and HF) were recorded. Results. Increased VS was found in 100% of TAVI patients with adverse event vs. 88.9% event-free TAVI patients (*p* = 0.116), and in 93.3% of AVR patients with event vs. 70.5% event-free (*p* = 0.061). Kaplan–Mayer free-survival curves at 1-year and 3-year FU were 90.5% vs. 97.1 % and 78% vs. 97.1% for patients with increased vs. lower VS. (*p* = 0.014). In univariate Cox analysis, elevated VS (HR 7.97, *p* = 0.04) and age (HR 1.05, *p* = 0.024) were associated with risk of adverse outcomes; however, both failed in Cox multivariable analysis. Conclusions. Vascular stiffness is associated with outcome after DAS intervention. However, it cannot be used as an independent outcome predictor.

## 1. Introduction

Resistive (RI) and pulsatile index (PI) are parameters corresponding to vascular stiffness (VS) which have been investigated in various clinical conditions, including renovascular and coronary atherosclerotic disease, hypertension, diabetes, and heart failure [1,2]. Vascular stiffness is a potential predictor of all-cause mortality, including cardiovascular mortality [3].

According to epidemiological data, even in young patients with increased VS, there is an increased risk of cardiovascular events, which carries with it a higher mortality rate [4]. Development and progression of degenerative aortic valve stenosis (DAS) is driven by similar factors to those of VS, including aging, atherosclerosis, inflammation, fibrosis, calcification processes, and genetic susceptibility [5,6,7]. In addition, the data behind genetic predispositions in patients with ischemic heart disease have proven to favor its significant importance in progression of VS [7].

However, it is still unclear whether RI and PI might predict the incidence of cardiovascular death (CVD) and heart failure (HF) episodes following transcatheter (TAVI) or surgical (AVR) intervention on DAS [8].

Therefore, in the present study, we aimed to assess whether VS is associated with outcomes in post-intervention DAS patients.

## 2. Materials and Methods

### 2.1. Study Population

The study group comprised 246 consecutive patients with severe symptomatic DAS (aortic valve area, AVA < 1.0 cm^2^) referred for surgical or interventional treatment. From this group, 119 patients underwent transcatheter aortic valve implantation (TAVI), while 127 patients underwent surgical aortic valve replacement (AVR). Afterwards, patients were followed up for 36 months for the composite endpoint: CVD and HF episodes requiring hospital readmission.

Subjects were eligible if they (1) had preserved left ventricular ejection fraction (LVEF), (2) had never been diagnosed with stroke or transient ischemic attack (TIA), and (3) were ≥40 years of age. The exclusion criteria for both study groups included significant stenosis of any carotid or vertebral artery (exceeding 50% lumen reduction), persistent atrial fibrillation or other severe arrhythmia, significant concomitant valvular diseases, ongoing or recent myocardial infarction (<3 months), hemodynamic instability (NYHA class IV or acute heart failure), aortic dissection, and lack of informed consent. 

Prevalence of cardiovascular risk factors including age, sex, hypertension, diabetes, and dyslipidemia was evaluated in compliance with guidelines of the European Society of Cardiology [9,10].

Carotid and vertebral arterial compliance parameters (RI and PI) of vascular stiffness indices and echocardiographic parameters of DAS were assessed. All measurements were done before final Heart Team qualification and performed by sonographers blinded to the subjects’ characteristics.

The study protocol was consistent with the requirements of the Helsinki Declaration and approved by the local Institutional Ethics Committee. All subjects gave their informed consent for participation in the study.

### 2.2. Echocardiographic Study

All patients underwent a complete echocardiographic study in compliance with guidelines of the European Association of Cardiovascular Imaging [11]. Peak velocity and mean gradient across the aortic valve, AVA, and LVEF were assessed in all subjects.

### 2.3. Arterial Compliance Assessment

High-resolution B-Mode, color Doppler, and pulse Doppler ultrasonography of both carotid and vertebral arteries were performed with an ultrasound machine (TOSHIBA APLIO 450) equipped with a linear-array 5–10 MHz transducer on a patient lying in the supine position with head tilted slightly backward. Examinations were performed by experienced sonographers who were blinded to the subject’s characteristics. Data comprised bilateral recording of peak systolic (PSV) and end diastolic velocities (EDV) measured within 1.0 to 1.5 cm of the proximal segment of the internal carotid artery and proximal V2 segment of the vertebral artery.

The averaged values of RI and PI from all assessed segments were calculated for each patient in accordance with the following equations: Resistive Index (RI) = [PSV − EDV/PSV], and Pulsatile Index (PI) = PSV − EDV/[(PSV + 2 × EDV)/3].

Frequencies of high RI (equal to 0.7 or higher) and high PI (equal to 1.3 or higher) from carotid and vertebral arteries were assessed [12,13].

### 2.4. Follow-Up Period

During an observation period of up to 36 months, the incidences of CVD and HF episodes were recorded. Cardiovascular disease was defined as fatal ischemic stroke, fatal myocardial infarction, fatal acute heart failure episode, or other CVD (i.e., any sudden or unexpected death unless proven as non-cardiovascular on autopsy). Heart failure episodes were defined as hospitalization for newly diagnosed or exacerbated congestive heart failure requiring administration of intravenous diuretics and/or vasoactive drugs (dopamine, dobutamine, epinephrine, or norepinephrine).

The final follow-up (FU) visit was conducted via telephone with the patient or an appointed family member. For all patients, data regarding patient vital status were obtained from the national health registry at the closing database.

### 2.5. Statistical Analysis

Data are presented as mean ± standard deviation or median (interquartile range) for continuous variables and as proportions for categorical variables. Differences between mean values were verified using the Student’s *t* test and analysis of variance (ANOVA) test, while frequencies were compared using the chi squared test for independence, as appropriate. Normal distribution of the studied variables was determined using the Shapiro–Wilk test. 

We assessed incidence of CVD and HF events in groups classified by high versus low PI and RI using the univariate Cox model, followed by the multivariable age-adjusted Cox models, with PI ≥ 1.3 and RI ≥ 0.7 as references [12,13]. We included age, sex, diabetes mellitus, hypertension, hyperlipidemia, previous myocardial infarction (MI), previous percutaneous coronary intervention (PCI), previous coronary artery bypass graft (CABG), chronic obstructive pulmonary disease (COPD), lower extremity artery disease (LEAD), LVEF, and pre-interventional AVA as factors which are potentially associated with the composite endpoint. 

Results of the multivariate Cox proportional hazards analysis were expressed as hazard ratio (HR) and 95% confidence interval (CI). A two-sided value of *p* < 0.05 was considered statistically significant. The Kaplan-Mayer survival curves were constructed for groups with high vs. low VS. Statistical analyses were performed with Statistica version 13.3 software (TIBCO Software, Palo Alto, CA, USA) and with R Statistic Language 3.6.3 (R-Core Team, Vienna, Austria) [14].

## 3. Results

Out of 249 initially screened patients with severe DAS, 246 were eligible for follow-up evaluation. Three patients died from perioperative complications: 2 in the AVR and 1 in the TAVI groups.

Successful AVR was performed in 127 patients having a mean age of 69.3 ± 7.2 years (range: 53–86), including 75 (59.1%) females. Successful TAVI was performed in 119 patients having a mean age of 80.5 ± 5.8 years (range: 58–88), including 85 (71.4%) females. 

The distribution of cardiovascular risk factors, including hyperlipidemia (*p* = 0.346), type 2 diabetes mellitus (*p* = 0.748), arterial hypertension (*p* = 0.292), history of previous MI (*p* = 0.833), and previous coronary interventions were similar between the AVR and the TAVI groups. Of note, patients referred for TAVI were older (*p* < 0.001) and more often were female (*p* = 0.042).

Patients with DAS referred for TAVI more frequently presented with symptoms corresponding to class 3 according to the New York Heart Association functional class (NYHA) when compared to AVR group (64.7% vs. 20.5%; *p* < 0.001).

All echocardiographic DAS parameters, as well as LVEF, were similar in both groups. 

Detailed study group characteristics are presented in Table 1.

Increased RI ≥ 0.7 and PI ≥ 1.3 were found in 91.6% of DAS patients in the TAVI group vs. 78.2% in the AVR group (*p* < 0.001) (Table 1).

After each valvular intervention, during a mean FU period of 29.3 ± 10.4 months, the composite endpoint occurred in 29 of 119 (24.4%) TAVI patients, including CVD in 21 (17.7%) and non-fatal HF episodes in 8 (6.7%) patients. In AVR patients, the composite endpoint occurred in 15 of 127 (11.8%) patients, including CVD in 7 (5.5%) and non-fatal HF episodes in 8 (6.3%) patients. A detailed comparison of patients with adverse events in TAVI and AVR groups is presented in Table 2.

Among patients with the composite endpoint compared to event-free patients, increased VS parameters (RI ≥ 0.7 and PI ≥ 1.3) were found in 29/29 (100%) vs. 80/90 (88.9%) patients in the TAVI group (*p* = 0.116), and in 14/15 (93.3%) vs. 79/112 (70.5%) in the AVR group (*p* = 0.061). In the entire study group (AVR plus TAVI), patients with increased VS more frequently suffered from a cardiovascular event when compared to patients with lower VS values (*p* = 0.011).

However, there was a large overlap of median and interquartile RI and PI values between event vs. event-free groups (Figure 1).

In the entire study group, Kaplan-Mayer free-survival curves at 1-year and 3-year FU were 90.5% vs. 97.1% and 78% vs. 97.1% for patients with increased VS compared to patients with lower RI and PI values (*p* = 0.014). Additionally, when TAVI and AVR groups were analyzed separately, patients with increased VS had lower free-survival curves when compared to patients with normal RI and PI values; however, this did not reach the level of statistical significance (Figure 2).

In univariate Cox analysis, factors potentially associated with increased risk of adverse outcomes included elevated VS (HR 7.97, 95% CI 1.10 to 57.9; *p* = 0.04), age (HR 1.05, 95% CI 1.01 to 1.09; *p* = 0.024), female gender (HR 1.90, 95% CI 0.94 to 3.85, *p* = 0.074), LEAD (HR 1.76, 95% CI 0.91 to 3.42; *p* = 0.094), and NYHA class III (HR 1.73, 95% CI 0.96 to 3.13; *p* = 0.069) (Table 3).

In multivariate Cox proportional hazard analysis, only LEAD (HR 2.22, 95% CI 1.12 to 4.39; *p* = 0.023) showed associations with risk of an adverse event, while increased VS failed to show an independent value (HR 7.12, 95% CI 0.97 to 52.5; *p* = 0.054) (Table 3).

## 4. Discussion

Our results support the hypothesis that high VS may be associated with risk of CVD and HF episodes in patients who underwent intervention for DAS, and that this risk can be elevated long after the intervention. Our findings are in line with studies by Makkar et al. and Mistiaen et al., who observed fatal cardiovascular events in patients with DAS and preserved LVEF, despite treatment of the valve [15,16].

Interestingly, the prognostic value of HF in patients with a preserved EF (HFpEF) in the TAVI population was presented by Seoudy et al. [17]. Importantly, the postulated multifactorial mechanisms of the HFpEF also may contribute to VS development and progression [18], while microvascular dysfunction underlies pathophysiological mechanisms of both VS and HF episodes despite a preserved systolic left ventricle contractility, constituting the main pathophysiological mechanism of recurrent HF episodes [19].

In the present study, preoperative values of RI ≥ 0.7 and PI ≥ 1.3, corresponding to increased VS, were associated with a 7.97-fold risk increase (*p* = 0.040) in univariate Cox proportional hazard analysis and a 7.12-fold risk increase (*p* = 0.054) in the occurrence of the composite endpoint in multivariate analysis. Similarly, Saeed et al. showed that event-free survival was significantly lower in patients with PWV ≥ 10 m/s when compared to those with lower PWV (*p* = 0.015); however, they observed an impact of PWV on all-cause mortality only in univariate Cox analysis (HR 1.80, 95% CI 1.14 to 2.83; *p* = 0.012) and not in multivariate analysis (HR 0.91, 95% CI 0.48 to 1.74, *p* = 0.778) [20].

In patients who underwent AVR for symptomatic DAS, increased left ventricular filling pressures were associated with cardiovascular mortality after AVR [21].

Similarly, in TAVI patients, VS is proposed as an important risk factor for adverse outcomes [22,23].

Tanaka et al. assessed the impact of pre-procedural brachial-ankle pulse wave velocity (PWV) on 1-year post-TAVI adverse outcomes in a group of 161 patients with severe DAS [22]. In the group with increased PWV, the incidence of all-cause death and re-hospitalization related to HF episodes was 3.42-fold greater (95% CI 1.62 to 7.85; *p* = 0.002) when compared to that of patients with lower PWV values [22]. Broyd et al. indicated an optimum cut-off for PWV higher than 11 m/s to be the only predictor of 1-year mortality following TAVI in 186 patients (OR 3.57, 95% CI 1.36–9.42, *p* = 0.01) associated with survival (log-rank *p* = 0.04) [23]. In line with these studies, our results indicate an important role for VS in the prediction of event-free survival at 1-year and 3-year FU, which were 90.5% vs. 97.1% and 78% vs. 97.1% for patients with increased VS when compared to patients with lower RI and PI values (*p* = 0.014).

Some researchers have investigated further, comparing VS parameters after DAS intervention. Musa et al. compared the impact of TAVI and AVR on VS as measured with PWV. They found that there was a further significant increase in PWV parameters following AVR at the 6-month FU, while in the TAVI arm, the postprocedural PWV increase did not reach the level of statistical significance [24]. However, in a TAVI population, Terentes-Printzios et al. showed that the arterial system exhibited increased stiffness in response to acute relief of the obstruction following intervention, which was retained in the long term [25]. Of note, in this high-risk subset of patients, such as patients referred for TAVI, the intervention on the valve has a beneficial effect on supra-aortic artery flow parameters during the orthostatic stress test, resulting from the alleviated obstruction to cerebral in-flow [26]. On the other hand, Cantürk et al. did not observe a significant change in PWV values following AVR [27].

In our study, we did not find a relationship between pre-interventional VS values and the NYHA class symptoms to the support findings of Kidher et al., which showed that PWV was an independent predictor of NYHA class pre-operatively (OR 8.3, 95% CI 2.27 to 33.33) and post-operatively (OR 14.44, 95% CI 1.49 to 139.31) [28]. Of interest, the baseline NYHA class (OR 1.02, 95% CI 1.005 to 1.041, *p* = 0.041) may be an independent predictor of improvement in PWV following AVR [28].

Our study shows the limitations of using VS parameters in daily clinical practice. The main disadvantage in result interpretation is the large overlap in median RI and PI values between groups with adverse events when compared to those without (Figure 1). A potential explanation for this finding is the presence of multifactorial associations between VS, traditional, and non-traditional cardiovascular risk factors [13,29]. 

Therefore, more data are required from studies in a larger scaled population to determine the role of VS in predicting outcome following aortic valve interventions. Recently published data from the OCEAN Japanese multicenter registry including 2588 patients who underwent TAVI demonstrated that male sex, body mass index, Clinical Frailty Scale, atrial fibrillation, peripheral artery disease, prior cardiac surgery, serum albumin level, renal function, and presence of pulmonary disease were independent predictors of 1-year mortality following TAVI [30]. However, in the registry of Yamamoto et al., arterial compliance was not investigated at all. Similarly, our present study showed LEAD to be an independent risk factor for CVD and HF episodes following aortic valve intervention. In summary, the Heart Team is at the center of the decision-making process in patients with DAS [31,32,33]. The gathered experience indicates that a multidimensional and multidisciplinary pre-procedural work-up in patients with severe DAS, including a thorough assessment of coexisting disorders, results in an optimal treatment strategy and can be associated with a superior prognosis when compared to conservative medical management [31,32,33]. 

Accordingly, future studies are required to elucidate whether routine VS assessment should be incorporated as an additional parameter in this risk stratification model. 

Seoudy et al. showed the clinical importance of a potential role in routine assessment of patients with HFpEF using the novel diagnostics algorithm (HFA-PEFF score) among the DAS population [17], where the same VS advancement score could be beneficial for better patient monitoring and treatment.

## 5. Conclusions

Our data demonstrates that VS is common in patients with severe DAS. We have demonstrated that increased VS can be a predictor of post-procedure outcome. In patients with PI ≥ 1.3 or RI ≥ 0.7, there is an increased risk of cardiovascular death and heart failure episodes despite intervention on the aortic valve (AVR or TAVI). However, huge the large overlap of RI and PI values between patients with or without adverse events during follow-up may limit the clinical value of routine vascular stiffness assessment.

## Figures and Tables

**Figure 1 jcm-11-02078-f001:**
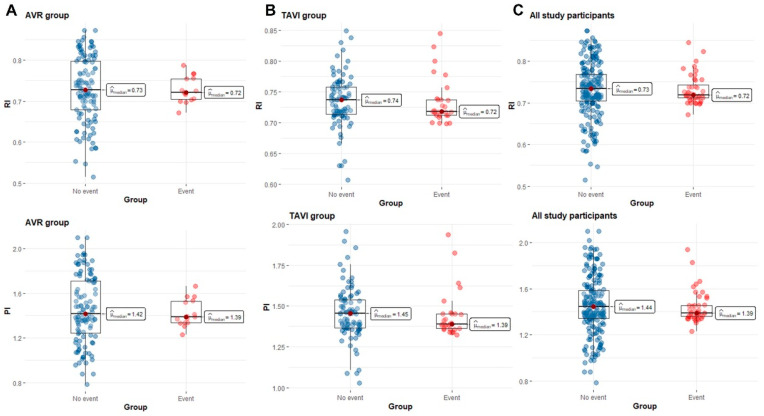
Median and interquartile range for mean RI and PI in patients with the composite endpoint and in patients without the endpoint. Panel (**A**), AVR group; Panel (**B**), TAVI group; Panel (**C**), all study participants (AVR and TAVI patients). Abbreviations: PI, pulsatile index; RI, resistive index.

**Figure 2 jcm-11-02078-f002:**
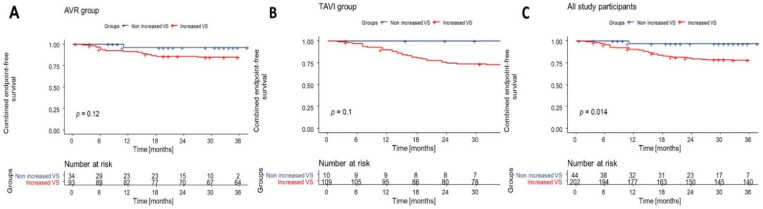
Kaplan-Meier survival curves showing time-to-event curves for 3-year cumulative survival to cardiovascular death and heart failure episodes dependent on increased vascular stiffness (defined as RI ≥ 0.7 and PI ≥ 1.3) compared to non-increased vascular stiffness. Panel (**A**), AVR group; Panel (**B**), TAVI group; Panel (**C**), all study participants (AVR and TAVI patients). Abbreviations: VS, vascular stiffness.

**Table 1 jcm-11-02078-t001:** Study group clinical data.

	AVR Group N = 127	TAVI Group N = 119	*p*-Value
**Demographic data**			
Age, years, (SD)	69.3 (7.2)	80.5 (5.8)	<0.001
Female, n (%)	75 (59.1)	85 (71.4)	0.042
Hypertension, n (%)	118 (92.9)	106 (89.1)	0.292
Diabetes, n (%)	43 (33.9)	38 (31.9)	0.748
Dyslipidemia, n (%)	124 (97.6)	118 (99.2)	0.346
Previous MI, n (%)	27 (21.3)	24 (20.2)	0.833
COPD, n (%)	9 (7.1)	14 (11.8)	0.208
Previous PCI, n (%)	31 (24.4)	34 (28.6)	0.459
Previous CABG, n (%)	7 (5.5)	9 (7.6)	0.514
LEAD, n (%)	24 (18.9)	19 (16.0)	0.545
NYHA III vs. I + II, n (%)	26 (20.5)	77 (64.7)	<0.001
**Echocardiographic data**			
Aortic valve area (cm^2^) ± SD	0.80 ± 0.2	0.70 ± 0.2	0.197
Peak aortic velocity (m/s) ± SD	4.76 ± 0.62	4.81 ± 0.68	0.193
Mean aortic gradient (mmHg) ± SD	52.6 ± 15.7	55.3 ± 19.2	0.596
Left ventricular ejection fraction (LVEF) (%) ± SD	61.8 ± 5.8	60.7 ± 7.0	0.368
**Vascular stiffness parameters**			
Resistive Index, median (Q1;Q3)	0.724 (0.685;0.784)	0.727 (0.714;0.756)	0.501
Pulsatile Index, median (Q1;Q3)	1.394 (1.272;1.650)	1.418 (1.364;1.527)	0.513
Resistive Index ≥ 0.7, n (%)	93 (78.2)	109 (91.6)	<0.001
Pulsatile Index ≥ 1.3, n (%)	93 (78.2)	109 (91.6)	<0.001

Abbreviations: AVR, aortic valve replacement; CABG, coronary artery bypass graft; COPD, chronic obstructive pulmonary disease; LEAD, lower extremities artery disease; MI, myocardial infarction; PCI, percutaneous coronary intervention; TAVI, transcatheter aortic valve implantation.

**Table 2 jcm-11-02078-t002:** Comparison of patients with composite endpoint and event-free patients in AVR and TAVI groups.

	AVR Group without Composite Endpoint N = 112	AVR Group with Composite Endpoint N = 15	*p*-Value	TAVI Group without Composite Endpoint N = 90	TAVI Group with Composite Endpoint N = 29	*p*-Value
**Demographic data**						
Age, years, (SD)	69.3 ± 7.2	68.9 ± 7.1	0.952	80.0 ± 5.9	82.3 ± 5.4	0.032
Female, n (%)	65 (58.0)	10 (66.7)	0.523	61 (67.8)	24 (82.8)	0.120
Hypertension, n (%)	103 (92.0)	15 (100)	0.598	78 (86.7)	21 (72.4)	0.349
Diabetes, n (%)	37 (33.0)	6 (40.0)	0.592	28 (31.1)	9 (31.0)	0.994
Dyslipidemia, n (%)	109 (97.3)	15 (100)	1.000	89 (98.9)	28 (96.6)	0.395
Previous MI, n (%)	8 (7.1)	1 (6.7)	0.946	12 (13.3)	2 (6.9)	0.349
COPD, n (%)	27 (24.1)	0 (0)	0.04	18 (20.0)	6 (20.7)	0.934
Previous PCI, n (%)	28 (25.0)	3 (20.0)	0.672	21 (23.3)	12 (41.4)	0.059
Previous CABG, n (%)	6 (5.3)	1 (6.7)	0.835	5 (5.6)	4 (13.8)	0.145
LEAD, n (%)	19 (17.0)	5 (33.3)	0.128	12 (13.3)	7 (24.1)	0.167
NYHA III vs. I + II, n(%)	21 (18.8)	5 (33.3)	0.189	59 (65.6)	18 (62.1)	0.733
**Echocardiographic data**						
Aortic valve area (cm^2^) ± SD	0.80 (0.20)	0.80 (0.27)	0.431	0.69 (0.19)	0.71 (0.22)	0.027
Peak aortic velocity (m/s) ± SD	4.79 (0.61)	4.54 (0.63)	0.296	4.79 (0.64)	4.84 (0.79)	0.064
Mean aortic gradient (mmHg) ± SD	53.0 (15.3)	49.5 (18.6)	0.284	54.7 (16.9)	58.1 (23.3)	0.091
LVEF (%) ± SD	61.7 (5.7)	62.8 (6.9)	0.341	60.7 (6.8)	60.9 (7.9)	0.460
**Vascular stiffness parameters**						
Resistive Index, median (Q1;Q3)	0.728 (0.678;0.797)	0.722 (0.705;0.755)	0.952	0.737 (0.715;0.758)	0.718 (0.712;0.737)	0.764
Pulsatile Index, median (Q1;Q3)	1.417 (1.242;1.713)	1.390 (1.335;1.526)	0.897	1.454 (1.368;1.538)	1.388 (1.361;1.451)	0.569
Resistive Index ≥ 0.7, n (%)	79 (70.5)	14 (93.3)	0.061	80 (88.9)	29 (100)	0.116
Pulsatile Index ≥ 1.3, n (%)	79 (70.5)	14 (93.3)	0.061	80 (88.9)	29 (100)	0.116

Abbreviations: AVR, aortic valve replacement; CABG, coronary artery bypass graft; COPD, chronic obstructive pulmonary disease; LEAD, lower extremities artery disease; MI, myocardial infarction; PCI, percutaneous coronary intervention; TAVI, transcatheter aortic valve implantation; VS, vascular stiffness.

**Table 3 jcm-11-02078-t003:** Univariate and multivariate Cox proportional hazard analysis presenting risk of cardiovascular death and heart failure episodes for increased VS (RI ≥ 0.7 and PI ≥ 1.3 in all study group participants.

	Univariate Cox Proportional Hazard Analysis			Multivariate Cox Proportional Hazard Analysis		
Variable	Hazard Ratio	95% Confidence Interval	*p*-Value	Hazard Ratio	95% Confidence Interval	*p*-Value
Age	1.05	1.01–1.09	0.024	1.02	0.97–1.06	0.420
Female gender	1.90	0.94–3.85	0.074	1.60	0.79–3.28	0.194
Hypertension	2.31	0.56–9.56	0.246			
Diabetes	1.10	0.59–2.04	0.774			
Dyslipidemia	0.50	0.07–3.61	0.488			
Previous MI	0.54	0.23–1.27	0.156			
COPD	0.69	0.21–2.23	0.534			
Previous PCI	1.41	0.75–2.62	0.284			
Previous CABG	1.94	0.76–4.92	0.163			
LEAD	1.76	0.91–3.42	0.094	2.22	1.12–4.39	0.023
Aortic valve area	0.90	0.20–3.99	0.893			
LVEF	1.00	0.96–1.05	0.943			
NYHA III vs. I + II	1.73	0.96–3.13	0.069	1.56	0.81–3.01	0.183
Increased VS (RI ≥ 0.7 and PI ≥ 1.3)	7.97	1.10–57.9	0.04	7.12	0.97–52.5	0.054

Abbreviations: CABG, coronary artery bypass graft; COPD, chronic obstructive pulmonary disease; LEAD, lower extremities artery disease; MI, myocardial infarction; LVEF, left ventricular ejection fraction; PCI, percutaneous coronary intervention; VS, vascular stiffness.

## Data Availability

The data presented in this study are available on request from the corresponding author. The data are not publicly available due to privacy.

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
