# Peer review of "Association of Increased Vascular Stiffness with Cardiovascular Death and Heart Failure Episodes Following Intervention on Symptomatic Degenerative Aortic Stenosis"

_jcm, 2022, doi:10.3390/jcm11082078_

Round 1

Reviewer 1 Report

The study is interesting and it focuses on the prognostic role of vascular stiffness in predicting the cardiovascula death and heart failure hospitalisation in patients with degenerative aortic stenosis. Methods are well described and results clearly presented. I have some suggestion to improve your maniscript:

1) Vascular stiffness is markedly influenced by the presence of cardiovascular risk factors and aging, which have high prevalence in patients with degenerative aortic stenosis. There is a growing interest in genetic predisposition to vascular stiffness and endothelial dysfunction which may predispose to cardiovascular disease, beyond risk factors presence (see  J Cardiovasc Dev Dis. 2021 Sep 18;8(9):116. doi: 10.3390/jcdd8090116). Do you think that a genetic predisposition may have an influence in determining the role vascular stiffness?

2) Patients included in your study have preserved ejection fraction. Are patients with acute decompensation HFrEF or HFpEF patients ? Vascular stiffness is associated with microvascular dysfunction, which is a main pathophysiological mechanism involved in HFpEF. Please add a brief discussion on this topic to justify the pathophysiological basis of HF in those patients (see ESC Heart Fail. 2022 Jan 29. doi: 10.1002/ehf2.13774; Kontogeorgos S, Thunström E, Pivodic A, Dahlström U, Fu M. Prognosis and outcome determinants after heart failure diagnosis in patients who underwent aortic valvular intervention. ESC Heart Fail. 2021 Aug;8(4):3237-3247. doi: 10.1002/ehf2.13451. Epub 2021 May 31; Int J Mol Sci. 2021 Jul 17;22(14):7650. doi: 10.3390/ijms22147650).  

3) add a brief discussion regarding how these results may impact clinical practice if confirmed on larger population. 

Author Response

We would like to thank the Reviewer for a careful evaluation of our study. We addressed all items with hope that our paper now is significantly improved and it will merit publication.

1) Vascular stiffness is markedly influenced by the presence of cardiovascular risk factors and aging, which have high prevalence in patients with degenerative aortic stenosis. There is a growing interest in genetic predisposition to vascular stiffness and endothelial dysfunction which may predispose to cardiovascular disease, beyond risk factors presence (see  J Cardiovasc Dev Dis. 2021 Sep 18;8(9):116. doi: 10.3390/jcdd8090116). Do you think that a genetic predisposition may have an influence in determining the role vascular stiffness? As you marked, genetic predisposition may have great influence also on vascular stiffness (VS) process. Probably the role of genetic is different at the beginning of VS development and different at more advances stages of VS. It could be great topic in the further research.

Answer: Thank you for paying attention to this issue. Of course, genetics is a major risk factor, now we have added a genetic susceptibility, as a risk factor for developing endothelial dysfunction and cardiovascular disease to the text:

‘Development and progression of degenerative aortic valve stenosis (DAS) is driven by similar factors to those of VS such as aging, atherosclerosis, inflammation, fibrosis, calcification processes and genetic susceptibility [5,6,7]. In addition, the data behind genetic predispositions in patients with ischemic heart disease have proven to favor its significant importance in progression of VS [7]. Severino, P.; D'Amato, A.; Prosperi, S.; Magnocavallo, M.; Mariani, M.V.; Netti, L.; Birtolo, L.I.; De Orchi, P.; Chimenti, C.; Maestrini, V.; et al. Potential Role of eNOS Genetic Variants in Ischemic Heart Disease Susceptibility and Clinical Presentation. J. Cardiovasc. Dev. Dis. 2021; 8: 116. doi: 10.3390/jcdd8090116.]

Indeed, the genetic allelic variants and polymorphisms would be of great interest for further research.

2) Patients included in your study have preserved ejection fraction. Are patients with acute decompensation HFrEF or HFpEF patients ? Vascular stiffness is associated with microvascular dysfunction, which is a main pathophysiological mechanism involved in HFpEF. Please add a brief discussion on this topic to justify the pathophysiological basis of HF in those patients (see

ESC Heart Fail. 2022 Jan 29. doi: 10.1002/ehf2.13774;

Kontogeorgos S, Thunström E, Pivodic A, Dahlström U, Fu M. Prognosis and outcome determinants after heart failure diagnosis in patients who underwent aortic valvular intervention. ESC Heart Fail. 2021 Aug;8(4):3237-3247. doi: 10.1002/ehf2.13451. Epub 2021 May 31;

Int J Mol Sci. 2021 Jul 17;22(14):7650. doi: 10.3390/ijms22147650).

Answer: We would like to thank the Reviewer for this important comment. We added a paragraph on this key issue to the Discussion:

‘Interestingly, the prognostic value of HF in patients with a preserved EF (HFpEF) in TAVI population was presented by Seoudy et al. [17]. Importantly, the postulated multifactorial mechanisms of the HFpEF also may contribute to VS development and progression [18]. While microvascular dysfunction underlies pathophysiological mechanisms of both VS and HF episodes despite a preserved systolic left ventricle contractility, constituting the main pathophysiological mechanism of recurrent HF episodes [19].

3) add a brief discussion regarding how these results may impact clinical practice if confirmed on larger population.

Answer: We have implemented in discussion the importance of HFpEF and potential great role of more accurate and active diagnostic of HFpEF in patients with aortic stenosis with preserve EF. 

‘Accordingly, future studies are required to elucidate whether routine VS assessment should be incorporated as an additional parameter in this risk stratification model. Seoudy et al. showed clinical importance of a potential role in routine assessment of patients with HFpEF using the novel diagnostics algorithm (HFA‐PEFF score) among DAS population [17], the same VS advancement score could be beneficial for better patients monitoring and treatment.’

Reviewer 2 Report

The authors aimed to assess whether VS might predict incidence of cardiovascular death (CVD) and heart failure (HF) episodes following intervention on degenerative aortic valve stenosis (DAS). In univariate Cox analysis, elevated VS (HR 7.97, p=0.04) and age (HR 1.05, p=0.024) were associated with risk of adverse outcomes, however both failed in Cox multivariable analysis. They concluded that vascular stiffness is associated with outcome after DAS intervention, but it cannot be used as an independent outcome predictor.

General comments

This is a manuscript addressing “Association of increased vascular stiffness with cardiovascular death and heart failure episodes following intervention on symptomatic degenerative aortic stenosis”. Although the discussion and conclusions drawn are supported by the results, some concerns need to be addressed.

Major concerns

  • In the multivariable analysis, the authors entered LEAD and VS. The two factors may relate each other and have the potential problem of “multicollinearity”. This might be the cause of the loss of significance in the multivariate analysis. Some options would be exclusion of LEAD, creating several models (e.g., age and VS; age, VS, and NYHA class).

Minor concerns

  • In fig. 1, the group of no-event has higher RI and PI. This may be a mistake.
  • The way of calculation of RI and PI should be explained more in detail (Line 84); these may be derived from PSV and/or EDV. The value of 0.7 and 1.3 would need references.
  • In fig. 2, the data for PI are missing.
  • The definition of VS according to PI and RI would be necessary in the methods.
  • The timepoint of the assessment of PI, RI, and echo-parameters would be necessary in the methods.

Author Response

We would like to thank the Reviewer for a careful evaluation of our study. We addressed all items with hope that our paper now is significantly improved and it will merit publication.

General comments

This is a manuscript addressing “Association of increased vascular stiffness with cardiovascular death and heart failure episodes following intervention on symptomatic degenerative aortic stenosis”. Although the discussion and conclusions drawn are supported by the results, some concerns need to be addressed.

Major concerns

  • In the multivariable analysis, the authors entered LEAD and VS. The two factors may relate each other and have the potential problem of “multicollinearity”. This might be the cause of the loss of significance in the multivariate analysis. Some options would be exclusion of LEAD, creating several models (e.g., age and VS; age, VS, and NYHA class).

Answer: Thank you for this comment. We have performed additional analyses, and there is no multicollinearity between LEAD and VS. On the contrary, when LEAD and VS were introduced together to the analysis, they both showed significance, suggesting that VS is not resulting from atherosclerosis alone, but it is more complex process, such as endothelial dysfunction, microvascular failures.

Minor concerns

  • In fig. 1, the group of no-event has higher RI and PI. This may be a mistake.

Answer: There is no mistake. We showed in results, a huge overlap in RI and PI median and interquartile values between event vs event-free group with lower median in event group. Importantly frequencies of the increased VS parameters were significantly higher for event positive cases in entire study group as well as in sub-groups (AVR and TAVI). We mentioned in discussion that potential explanation for this finding is the presence of multifactorial associations between VS, traditional, and non-traditional cardiovascular risk factors.

  • The way of calculation of RI and PI should be explained more in detail (Line 84); these may be derived from PSV and/or EDV.

Answer: Indeed, the measurements derived from the peak and end-diastolic velocities. Now, we have added the respective equations for RI and PI in the Material and Methods:

‘The averaged values of RI and PI according to the following equations: Resistive Index (RI) = [PSV – EDV/PSV], and Pulsatile Index (PI) = PSV – EDV/[(PSV + 2 x EDV)/3] from all assessed segments were calculated for each patient.’

  • The value of 0.7 and 1.3 would need references.

Answer: The cut-offs were adopted from the previous publications. We have added the following refs:

Frauchiger B, Schmid HP, Roedel C, Moosmann P, Staub D. Comparison of carotid arterial resistive indices with intima-media thickness as sonographic markers of atherosclerosis. Stroke. 2001 Apr;32(4):836-41. doi: 10.1161/01.str.32.4.836.

Baran, J.; Kleczynski, P.; Niewiara, Ł.; Podolec, J.; Badacz, R.; Gackowski, A.; Pieniazek, P.; Legutko, J.; Zmudka, K.; Przewłocki, T.; et al. Importance of Increased Arterial Resistance in Risk Prediction in Patients with Cardiovascular Risk Factors and Degenerative Aortic Stenosis. J. Clin. Med. 2021, 10, 2109

  • In fig. 2, the data for PI are missing.

Answer: Thank you very much for this comment. As, RI and PI are strongly correlated with each other, we based Fig.2 only on the RI. Now, we have changed figure showing increased vascular stiffness defined as RI ≥ 0.7 and PI ≥ 1.3 vs non-increased vascular stiffness

  • The definition of VS according to PI and RI would be necessary in the methods.

            Answer: Definition was updated in methodology.

‘The averaged values of RI and PI according to the following equations: Resistive Index (RI) = [PSV – EDV/PSV], and Pulsatile Index (PI) = PSV – EDV/[(PSV + 2 x EDV)/3] from all assessed segments were calculated for each patient. Frequencies of high RI (equal to 0.7 or higher) and high PI (equal to 1.3 or higher) from carotid and vertebral arteries were assessed [12,13]. ‘

  • The timepoint of the assessment of PI, RI, and echo-parameters would be necessary in the methods.

Answer: Both measurements (ultrasonography and echo) were done before Heart Team qualification and performed by sonographers blinded to subject’s characteristics.

‘Carotid and vertebral arterial compliance parameters (RI and PI) of vascular stiffness indices and echocardiographic parameters of DAS were assessed. All measurements were done before final Heart Team qualification and performed by sonographers blinded to subject’s characteristics.’

Round 2

Reviewer 1 Report

All the suggestions have been correctly satisfied 

Reviewer 2 Report

The authors have responded appropriately to my concerns, providing additional data.